# A COVID-19 DNA Vaccine Candidate Elicits Broadly Neutralizing Antibodies against Multiple SARS-CoV-2 Variants including the Currently Circulating Omicron BA.5, BF.7, BQ.1 and XBB

**DOI:** 10.3390/vaccines11040778

**Published:** 2023-03-31

**Authors:** Yuan Ding, Feng Fan, Xin Xu, Gan Zhao, Xin Zhang, Huiyun Zhao, Limei Wang, Bin Wang, Xiao-Ming Gao

**Affiliations:** Advaccine Biopharmaceutics (Suzhou) Co., Ltd., Suzhou 215123, China

**Keywords:** COVID-19, SARS-CoV-2, DNA vaccine, RBD chimera, Omicron

## Abstract

Waves of breakthrough infections by SARS-CoV-2 Omicron subvariants currently pose a global challenge to the control of the COVID-19 pandemic. We previously reported a pVAX1-based DNA vaccine candidate, pAD1002, that encodes a receptor-binding domain (RBD) chimera of SARS-CoV-1 and Omicron BA.1. In mouse and rabbit models, pAD1002 plasmid induced cross-neutralizing Abs against heterologous sarbecoviruses, including SARS-CoV-1 and SARS-CoV-2 wildtype, Delta and Omicron variants. However, these antisera failed to block the recent emerging Omicron subvariants BF.7 and BQ.1. To solve this problem, we replaced the BA.1 RBD-encoding DNA sequence in pAD1002 with that of BA.4/5. The resulting construct, namely pAD1016, elicited SARS-CoV-1 and SARS-CoV-2 RBD-specific IFN-γ^+^ cellular responses in BALB/c and C57BL/6 mice. More importantly, pAD1016 vaccination in mice, rabbits and pigs generated serum Abs capable of neutralizing pseudoviruses representing multiple SARS-CoV-2 Omicron subvariants including BA.2, BA.4/5, BF.7, BQ.1 and XBB. As a booster vaccine for inactivated SARS-CoV-2 virus preimmunization in mice, pAD1016 broadened the serum Ab neutralization spectrum to cover the Omicron BA.4/5, BF7 and BQ.1 subvariants. These preliminary data highlight the potential benefit of pAD1016 in eliciting neutralizing Abs against broad-spectrum Omicron subvariants in individuals previously vaccinated with inactivated prototype SARS-CoV-2 virus and suggests that pAD1016 is worthy of further translational study as a COVID-19 vaccine candidate.

## 1. Introduction

Effective vaccines against severe acute respiratory syndrome coronavirus 2 (SARS-CoV-2) are essential tools for controlling the ongoing pandemic coronavirus disease 2019 (COVID-19), which has caused more than 630 million infections with more than 6.5 million deaths worldwide since late 2019 [1]. To date, more than 30 first-generation vaccines based on the wildtype (WT) strain of SARS-CoV-2 and several second-generation vaccines based on SARS-CoV-2 variants of concerns (VOCs) have been approved or authorized for emergency use. These include inactivated virus, viral vector, recombinant subunit and nucleic acid vaccines [2]. Significantly decreased protective efficacies offered by the currently available vaccines against newly emerging variants were observed in clinical trials and real-world evidence studies of first-generation COVID-19 vaccines [2,3,4,5]. In the first half of 2022, Omicron subvariants BA.4 and BA.5 circulated globally and gradually substituted their predecessors BA.1 and BA.2 [5,6,7,8,9]. Mutations in the BA.4/5 spike (S) protein led to resistance for humoral immune responses induced by predecessor variants [10]. In recent months, new escape mutants Omicron BF.7, BQ1 and XBB began to circulate in different parts of the world [11]. Consequently, it is necessary to create novel vaccines able to provide broader-spectrum protection against the newly emerging Omicron VOCs.

The spike (S) protein of SARS-CoV-2 is the main target for COVID-19 vaccines. It contains the receptor-binding domain (RBD) responsible for human ACE2 (hACE2) receptor binding and mediating virus entry [12,13]. Neutralizing antibodies (NAbs) specific for RBD in the S1 region of the S protein play critical roles in COVID-19 protection [14,15]. Thus far, SARS-CoV-2 S protein-encoding mRNA vaccines have shown high protection efficacy in human populations [2]. However, mRNA vaccines require low-temperature conditions for storage and distribution. Moreover, worrying side-effects associated with mRNA administration in humans have been reported. DNA vaccines are considered an attractive alternative to conventional vaccines because they are relatively easy and inexpensive to produce, stable at room temperature, safe to use and able to stimulate robust cellular as well as humoral immunity in vivo [16,17]. Several groups have explored the possibility of developing various DNA vaccines against COVID-19 with promising results [18,19,20]. For example, COVID-eVax, an electroporated DNA vaccine candidate encoding the ancestral SARS-CoV-2 RBD, elicited protective responses in animal models [19]. pGX9501, a WT SARS-CoV-2 full-length (FL) S-protein-encoding electroporated DNA vaccine candidate, was able to elicit NAbs as well as IFN-γ^+^-CD4 and CD8 T cells against WT SARS-CoV-2 as well as Delta variant in volunteers aged between 18 and 60 years [20]. In 2021, ZyCoV-D, an S-protein-encoding DNA vaccine delivered by a needleless injector, was authorized for emergency use against COVID-19 in India [21]. We recently documented a COVID-19 DNA vaccine candidate, pAD1002, that encodes a secreted form of fusion RBD of SARS-CoV-1 (RBD^SARS^) and SARS-CoV-2 Omicron BA.1 (RBD^BA1^). When delivered by either microneedle array patch (MAP) or electroporation, the pAD1002 plasmid elicited high titer NAbs against heterologous sarbecoviruses including SARS-CoV-1, SARS-CoV-2 WT and its variants Beta, Delta and Omicron BA.1, BA.2, and, relatively poorly, BA.4/5 [22]. However, pAD1002 antisera from BALB/c mice and rabbits were unable to neutralize Omicron BQ.1 and BF.7 pseudoviruses [22]. In the present study, an adaptation construct, namely pAD1016, was generated by replacing the RBD^BA1^-encoding sequence in pAD1002 with that of BA.4/BA.5. Mouse antibodies (Abs) elicited by plasmid pAD1016, in either a two-dose DNA vaccination procedure, or a prime-boost scheme with the DNA vaccine as a booster dose, effectively cross-neutralized multiple Omicron subvariants, including BA.4/5, BF.7 and BQ.1. pAD1016-induced rabbit Abs effectively neutralized BA.2, BA.4/5, BF.7 and XBB pseudoviruses. Moreover, after immunization with pAD1016 twice, pigs produced serum Abs capable of neutralizing BQ.1.1 pseudovirus. Our RBD^SARS^ chimera approach paves the way for expedited vaccine development to catch up with the continually mutating SARS-CoV-2 virus.

## 2. Materials and Methods

### 2.1. DNA Vaccine Construction

Preparation of pVAX1-based plasmid pAD1002 was as previously described [22]. The cloning strategy and structural characteristics of pAD1016 are illustrated in Appendix A. The synthesis of cDNA encoding RBD of Omicron BA.5 was performed by GenScript, Nanjing, China. Optimization analysis of the cDNA sequences was performed using an in-house analytic tool, taking into account codon usage bias, GC content, mRNA secondary structure, cryptic splicing sites, premature poly(A) sites, internal *chi* sites and ribosomal binding sites, negative CpG islands, RNA instability motif (ARE), repeat sequences (direct repeat, reverse repeat and dyad repeat) and restriction sites that may interfere with cloning. The resulting synthesized and optimized cDNA was cloned into pAD1002, replacing the BA.1 RBD-encoding DNA fragment. Restriction enzyme analysis and DNA sequencing were performed to confirm the accuracy of construction. Plasmids were transformed into *E. Coli* strain HB101. Single colonies underwent expansion in one-liter flasks for culturing in LB broth. Plasmids were extracted, purified by MaxPure Plasmid EF Giga Kit (Magen, China), and dissolved in distilled water at 1 mg/mL final concentration. Purity of the plasmids was measured by an agarose gel electrophoresis and a UV detector at a range of 1.8–2.0 OD260 nm/280 nm. Endotoxin contamination in plasmid samples was below 30 EU/mg by the LAL test.

### 2.2. MAP-1016 Preparation

Dissolvable microneedle array patches (MAP) each laden with 20 μg pAD1016 DNA was fabricated as previously described [22]. For the immunization of rabbits, larger-sized MAPs (52 × 25 mm, 3000 microneedles per patch) laden with 500 μg pAD1016 DNA were prepared using the same procedure.

### 2.3. Inactivated Prototype SARS-CoV-2 Virus Vaccine

Inactivated WT SARS-CoV-2 vaccine was a gift from Sinovac Biotech Co. Ltd, Beijing, China.

### 2.4. Western Blot

HEK293 cells pre-plated in a 6-well plate were transiently transfected with 2.5 μg DNA vaccine plasmids with Hieff TransTM Liposomal Transfection Reagent (YEASEN, Shanghai, China). Two days later, the cells were pelleted and lysed in immunoprecipitation assay buffer. Cell lysates and supernatants were separated by SDS-PAGE and transferred to PVDF membranes. Immunoblotting was performed by using rabbit anti-RBD^WT^ primary Ab (Bioworld, Nanjing, China) diluted 1:1000 in 5% milk-0.05% PBS-Tween 20 and horseradish peroxidase (HRP)-labeled goat anti-rabbit IgG secondary Ab (BD Biosciences, San Diego, CA, USA). Chemiluminescence detection was performed with the ECL Prime Western Blotting System and acquired by the ChemiDoc Imaging System (Bio-Rad, Hercules, CA, USA).

### 2.5. qRT-PCR

SARS-CoV-2 RBD-specific quantitative reverse transcription-PCR (qRT-PCR) assays were performed by using a FastKing One Step Probe RT-qPCR kit (Tiangen Biotech, Shanghai, China) on a CFX96 Touch real-time PCR detection system (Bio-Rad, Hercules, CA, USA). Sequences of the 3′ and 5′ end probes are: AAGCTGAACGACCTGTGCTTCA and GGCAGCTTGTAGTTGTAG.

### 2.6. Animal Immunization

Female BALB/c and C57BL/6 mice (6–8 weeks of age) were purchased from Shanghai SLAC Laboratory Animal Co., Ltd. (Shanghai, China) and maintained under specific pathogen-free (SPF) conditions at the animal facilities of Advaccine Biologics (Suzhou) Co. New Zealand white rabbits, purchased from Shanghai Somglian Experimental Animal Company (Shanghai, China), were housed in the Grade I animal facilities of Advaccine Biologics Co. (Suzhou, China). All animal experiments were performed in compliance with the recommendations in the Guide for the Care and Use of Laboratory Animals of the Ministry of Science and Technology Ethics Committee and approved (document No. 2021070102) by the Ethics Committees of the company.

EP-assisted DNA immunization was performed in mouse and rabbit quadriceps injected with 20 μg (for mice) or 0.5 mg (for rabbits) DNA (1 mg/mL in 30 μL SSC), followed by EP using Inovio CELLECTRA^®^2000 and electrode (Inovio, San Diego, CA, USA) with two sets of pulses with 0.2 amp constant current. All intramuscular (IM)+EP-delivered vaccines were primed on day 0 and boosted on day 14 unless otherwise indicated. Blood samples were collected on days 0, 14, 21 and/or 28.

### 2.7. Enzyme-Linked Immunosorbent Assay

Antibody titration was performed on sera obtained by retro-orbital bleeding from mice or venous bleeding from rabbit’s ears. The ELISA plates were functionalized by coating with the recombinant RBD of wildtype SARS-CoV-2 virus (SinoBiological, Beijing, China) at 1 μg/mL, incubated 18 h at 4 °C and subsequently blocked with 3% BSA-0.05% Tween 20-PBS (PBST) for 1 h at room temperature. Serial diluted serum samples were then added in triplicate wells, and the plates were incubated for 1 h at room temperature. After a double wash with PBST, horseradish peroxidase (HRP)-conjugated Ab against murine (Abcam, ab6789, Cambridge, UK, 1/2000 diluted), or rabbit (GenScript, Nanjing, China, A00098, 1:2000 diluted) IgG was added and then developed with 3,3′,5,5′-tetramethylbenzidine (TMB) substrate (Coolaber, Beijing, China). The reaction was stopped with 2 M of H_2_SO_4_, and the absorbance measured at 450 nm and reference 620 nm using a microplate reader (TECAN, Männedorf, Switzerland).

### 2.8. Neutralization Antibody Detection

The pseudovirus microneutralization assay was performed to measure neutralizing antibody levels against prototype SARS-CoV-2 and variants. Pseudovirus stocks of SARS-CoV-2 prototype, variants Delta, BA.2, BA.4/5, BF.7, BQ.1 and XBB were purchased from Gobond Testing Technology, and were aliquoted for storage at −80 °C. hACE2 stable expressing HEK293T cells (prepared in our lab) were used as target cells plated at 10,000 cells/well. SARS-CoV-2 pseudoviruses (50 μL) were incubated with heat-inactivated (56 °C for 30 min) and 1/3 serial diluted mouse sera (50 μL) for 90 min at room temperature; then, the sera-pseudovirus mixtures were added to hACE2-HEK293T cells and allowed to incubate in a standard incubator 37% humidity, 5% CO_2_ for 72 h. The cells were then lysed using Bright-Glo™ Luciferase Assay (Promega Corporation, Madison, WI, USA), and relative luminance unit (RLU) was measured using an automated luminometer. Fifty percent pseudovirus neutralization titer (pVNT50) was determined by fitting nonlinear regression curves using GraphPad Prism and calculating the reciprocal of the serum dilution required for 50% neutralization of infection. These assays were performed in a BSL-2 facility of Advaccine. Pseudovirus neutralization results were verified using Vero cells by Gobond Testing Technology.

### 2.9. Molecular Structure AI Modeling

AlphaFold2 was used for structure prediction with the required homology modeling databases running on ColabFold. The pLDDT plots generated and the obtained structures were further visualized by PyMol 2.4.

### 2.10. ELISpots

Spleens and draining lymph nodes (LN) from immunized mice were collected and used to prepare single cell suspension in RPMI-1640 medium supplemented with 10% FBS and penicillin/streptomycin. ELISpot was performed using mouse IFN-γ ELISpot PLUS kits (MABTECH, Cincinnati, OH, USA) according to the manufacturer’s protocol. Briefly, 5 × 10^5^ freshly prepared mouse splenocytes, or LN cells, were plated into each well and stimulated for 20 h with pooled overlapping 15-mer peptides (10 μg/mL) covering respective SARS-CoV-1 or SARS-CoV-2 Omicron BA4/5 RBDs at 37 °C in a 5% CO_2_ incubator. PMA(phorbol-12-myristate-13-acetate)/Ionomycin was used for positive controls. The plates were processed in turn with a biotinylated detection antibody. Spots were scanned and quantified using AID ImmunoSpot reader (AID, GER). IFN-γ spot-forming units were calculated and expressed as SFUs per million cells.

### 2.11. Statistics

Statistical analyses were performed with GraphPad Prism software version 9 (GraphPad Software, Boston, MA, USA). Error bars indicate the standard error of the mean (SEM). We used Mann–Whitney *t*-tests and two-way ANOVA tests to analyze experiments with multiple groups and two independent variables. Significance is indicated as follows: * *p* < 0.05; ** *p* < 0.01. Comparisons are not statistically significant unless indicated.

## 3. Results

### 3.1. Preparation of DNA Construct Encoding RBD Chimera of SARS-CoV-1 and SARS-CoV-2 Omicron BA.4/5

To overcome the inability of DNA vaccine candidate pAD1002 to generate NAbs against the most recent emerging Omicron subvariants, an adaptation construct, pAD1016, was prepared by replacing the RBD^BA1^-encoding DNA sequence in pAD1002 with that of BA.4/5 (Figure 1A and Appendix A). As with plasmid pAD1002, pAD1016 was expressed in high levels after transfection into HEK293T cells, as evidenced by qPCR and Western blotting (Figure 1B,C). The secreted recombinant RBD^SARS/BA5^ polypeptide was readily detectable by ELISAs in culture supernatant of the transfectant cells (Figure 1D). Intradermal inoculation of naked pAD1016 (2 doses, 20 μg/dose with a fortnight interval) led to strong RBD-specific (RBD^WT^-cross-binding) IgG responses in BALB/c and C57BL/6 mice (Figure 1E,F). EP further enhanced the immunological responses to pAD1016 i.m. immunization in mice (Figure 1E,F). Additionally, human ACE2-transgenic K18 mice responded to pAD1016 immunization by producing high titer RBD-cross-binding serological IgG (Appendix A). Together these data confirm that pAD1016 inherited the strong intracellular expression capability and robust in vivo immunogenicity of its predecessor pAD1002.

### 3.2. Cellular Responses Elicited by pAD1016 in Mice

Compared to inactivated virus or subunit viral protein vaccines, nucleic acid vaccines are particularly powerful in generating MHC-I-restricted CD8^+^ cytotoxic T lymphocytes (CTL), known to play pivotal roles in protection against viral infections in vivo [17]. SARS-CoV-2-virus-specific CTL responses have been found to be associated with milder situations in acute and convalescent COVID-19 patients [23]. We recently showed that plasmid pAD1002 was able to induce RBD-specific CTL responses in mice, evidenced by pAD1002-induced expansion of CD8^+^ T cells bearing T cell receptors (TCRs) specific for a H-2D^b^-restricted predominant CTL epitope (the “S” epitope) in the RBD of SARS-CoV-2 [22]. In the present study, pAD1016 was compared with pAD1002 and pVAX1 (as a negative control) for the induction of cellular responses in C57BL/6 mice. ELISpot assay results indicated that pAD1016 was comparable to pAD1002 in eliciting IFN-γ^+^ cells responsive to SARS-CoV-1 and SARS-CoV-2 BA.1/2 RBD peptides (Figure 2), further endorsing the “pAD1002-like” strong immunogenicity of pAD1016 in vivo.

### 3.3. Neutralizing Abs induced by pAD1016 in Mice and Rabbits

The generation of NAbs is known to be crucial for protecting people from virus infection. NAb levels are highly predictive of immune protection from symptomatic SARS-CoV-2 infection in humans [14,15]. To gain insight on pAD10160-generated NAbs in vivo, antisera from pAD1016-immunized C57BL/6 mice and rabbits were tested for their ability to block the mimic infection of ACE2-transgenic HEK293T cells by pseudoviruses displaying the S protein of the SARS-CoV-2 wildtype virus or variants. As shown in Figure 3A, the pAD1016-generated mouse antisera strongly neutralized the pseudoviruses of Omicron BA.2 and BA.4/5 (mean NT50 titers 1623 and 3024, respectively) but relatively poorly against the pseudoviruses of SARS-CoV-2 WT and Delta variant (mean NT50 titers 130–160). Consistently, pAD1016-immunized hACE2-transgenic K18 mice produced antisera showing high titer cross-binding to recombinant RBD^WT^ and strong neutralization of BA.4/5 pseudovirus. This is contrasted by pAD1002-generated Abs, which were strong binders to recombinant RBD^WT^ in ELISAs but very poor blockers of BA.4/5 pseudovirus in neutralization assays (Appendix A). Furthermore, pAD1016-generated rabbit antisera strongly neutralized Omicron subvariants BA.2, BA.4/5 and BF.7 but scored poorly against the SARS-CoV-2 WT and Delta variant (mean NT50 titers 500–1100 vs. 40–60) (Figure 3B). It was previously shown that MAP-mediated intradermal delivery of pAD1002 DNA in rabbits generated serum Abs capable of neutralizing the Omicron BA.2 but not the BF.7 subvariants [22]. Here, when rabbits were intradermally administered with two doses of pAD1016 DNA using MAP-1016 (2 × 500 μg DNA/patch/dose), the resulting antisera effectively neutralized Omicron subvariants BF.7 and XBB (Figure 3C). Moreover, pAD1016 was able to induce NAbs against Omicron subvariants BF.7 and BQ.1.1 in pigs (Appendix A). Taken together, these results reveal a clear neutralization spectrum shift of pAD1016 antisera towards recent emerging Omicron subvariants compared to that elicited by pAD1002 in animals of different species.

### 3.4. Plasmid pAD1016 as a Booster Dose to Inactivated Prototype SARS-CoV-2 Virus Pre-Vaccination

Given that most people in China have been vaccinated with inactivated prototype SARS-CoV-2 virus vaccines in the last two years, the most likely use of future COVID-19 DNA vaccines in this country would be as a heterologous booster to inactivated vaccine preimmunizations. We therefore set out to address the question of whether a heterologous prime-boost scheme combining an inactivated prototype SARS-CoV-2 virus vaccine and a pAD1016 DNA vaccine would generate good protective immunity against the newly emerging Omicron subvariants in vivo. C57BL/6 mice were given two doses of the inactivated prototype virus vaccine (60 units/dose) followed by a booster dose of either pAD1016 (20 μg/dose, IM+EP delivery) or the inactivated virus vaccine. Figure 4A–C reveal that three doses of inactivated virus vaccine generated cross-neutralizing Abs against Omicron BA.4/5 but not BA.7 and BQ.1. Interestingly, however, the heterologous boost using pAD1016 after two doses of inactivated virus vaccine immunization led to the production of serological NAbs against Omicron BA.4/5 and BF.7 as well as BQ.1. As shown in Figure 4D, two doses of inactivated virus immunization generated high titer NAbs against the SARS-CoV-2 Delta variant, and a booster dose of the same vaccine further enhanced the level of such NAbs. Interestingly, pAD1016 booster immunization also significantly enhanced Delta-specific NAb titers, highlighting the potential benefit of pAD1016 in eliciting broadly protective Abs in individuals previously given the inactivated prototype SARS-CoV-2 virus vaccination.

## 4. Discussion

Building upon our recent work on DNA vaccine candidate pAD1002 encoding RBD^SARS/BA1^ chimera, here we demonstrate the modification of pAD1002 by replacing the RBD^BA1^-encoding sequence with that of RBD^BA5^. The resulting plasmid, pAD1016, generated Abs capable of neutralizing multiple Omicron subvariants including BA.4/5, BF.7, BQ.1 and XBB in mice and rabbits, thereby overcoming the inability of pAD1002 in covering the recent emerging Omicron subvariants. In addition, pAD1016 was able to induce NAbs against Omicron subvariants BF.7 and BQ.1.1 in pigs (Appendix A C,D). More importantly, as a booster dose to inactivated virus pre-vaccination, pAD1016 generated NAbs against recent emerging Omicron subvariants. These results provide proof-of-concept evidence for our “fast DNA vaccine adaptation” strategy, which may be of great value in the control of the long-persisting COVID-19 pandemic caused by the constantly mutating SARS-CoV-2 virus.

Recombinant RBD heterodimers of SARS-CoV-2 VOCs have been shown to be strong immunogens capable of eliciting robust cross-protective immune responses in various model animals [24,25]. We previously compared IgG responses induced by IM-inoculated RBD^Beta/BA1^-encoding construct pAD1003 or RBD^SARS^ chimera-encoding plasmids pAD1002 and pADV131 in the absence of adjuvant or EP. In those experiments, pAD1002 and pADV131 exhibited much stronger immunogenicity than pAD1003 in animal models of different genetic backgrounds [22]. In this study, we found that direct ID injection of naked pAD1016 DNA without EP assistance also generated a relatively strong IgG response in BALB/c and C57BL/6 mice (Figure 1E,F). These results together strongly support the idea that RBD^SARS^ possesses immuno-potentiating capability in vivo. The molecular mechanisms for this phenomenon are not yet fully understood. Figure 5 shows the results of molecular structure AI modeling on polypeptides encoded by pAD1002, pAD1003, pAD131 and pAD1016. The pAD1003 polypeptide (Beta-BA1 RBD chimera) adopted a “bilateral lung-like” structure previously observed in prototype–prototype homodimer and prototype–Beta SARS-CoV-2 RBD chimera by Xu et al. [24]. By contrast, all three RBD^SARS^-containing polypeptides adopted a “free-moving” structure in which RBD^SARS^ and the pairing RBD of SARS-CoV-2 variants were kept apart from each other in solution. It is likely that RBD^SARS^ could not form a stable tertiary dimeric structure with the RBD of the SARS-CoV-2 virus under physiological conditions. Presumably, the free-moving structure of the RBD^SARS/SARS-CoV-2^ chimera would allow the exposure of more binding sites in RBDs for NAbs, hence improving the immunogenicity of heterodimeric RBD polypeptides or their encoding DNA constructs. It is worth noting that the BNT162b2 mRNA vaccine generated pan-sarbecovirus NAbs in SARS-CoV survivors, suggesting that SARS-CoV-induced immunological memory cells could help the production of broadly cross-reactive NAbs against SARS-CoV-2 variants [26]. This may help to explain why the combination of RBDs from pre-emergent SARS-CoV-1 and SARS-CoV-2 variants would generate stronger immunogens against SARS-related viruses. In any case, the RBD^SARS^-encoding sequence is a useful component in the future design of DNA or recombinant subunit vaccines.

Several groups have explored the possibility of developing S-protein- or RBD-encoding DNA vaccines against COVID-19, delivered by EP or a needless injection [18,19,20,21]. Consistent with our recent work showing that MAPs can effectively substitute EP and a needless injection for DNA vaccine delivery in animal models of different genetic backgrounds [22], MAP-based pAD1016 also exhibited strong immunogenicity in generating strong adaptive immune responses in mice and rabbits. MAP-mediated intradermal delivery of a DNA vaccine provides a new choice of vaccine administration in the future. The skin dermis layers are ideal sites for vaccination, particularly DNA vaccination. MAP-delivered plasmid DNA is intradermally expressed for about 2 weeks after administration in mice [22]. The skin layers contain abundant antigen-presenting cells (APCs) such as Langerhans cells and dendritic cells (DCs) that play important roles in inducing adaptive immunity [27,28,29]. Vaccine DNA unloaded from an applied 1 cm^2^ skin patch will directly reach some 100,000 APCs in the dermis layer, which could uptake DNA plasmids and then migrate them to draining LNs as matured APCs expressing the encoded antigen, thereby triggering strong adaptive immunological responses [30].

In conclusion, the pAD1002-based COVID-19 DNA vaccine can be relatively easily modified, by replacing the SARS-CoV-2 RBD-encoding sequence, to cover the newly emerging SARS-CoV-2 VOCs. The circulating mutant-targeted (adaptation) replacement approach in DNA vaccine development merits further translational studies.

## Figures and Tables

**Figure 1 vaccines-11-00778-f001:**
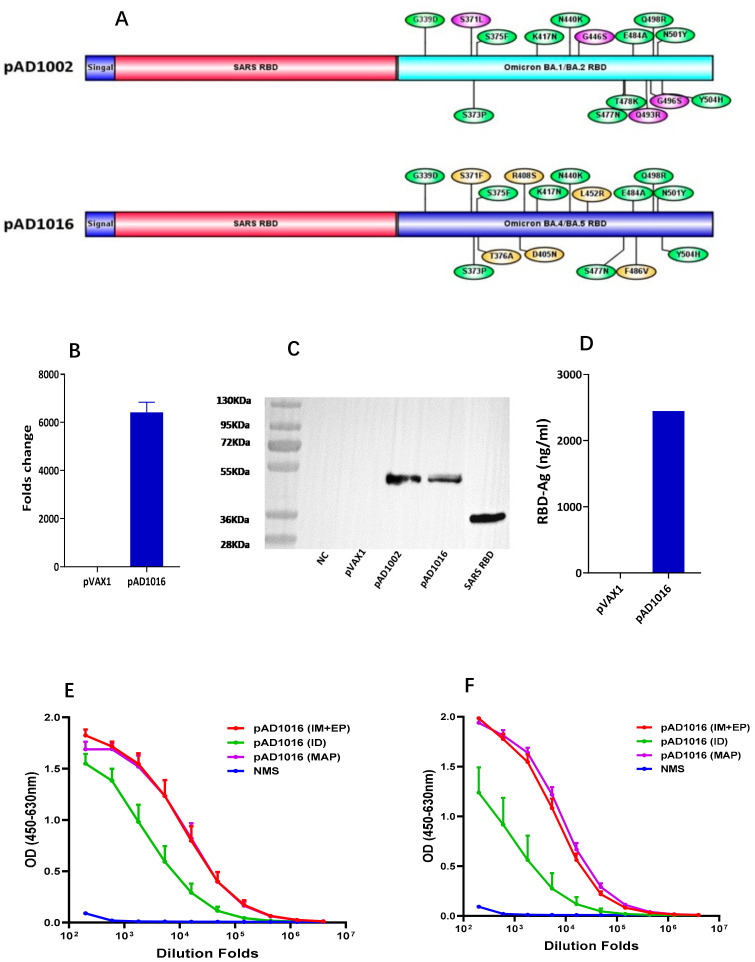
Construction of pAD1016. (**A**) Schematic diagram comparing the primary structure of secreted polypeptides encoded by pVAX1-based vaccine candidates pAD1002 and pAD1016. (**B**) qPCR detection of RBD mRNA transcripts in HEK293T cells transiently transfected with pVAX1 or pAD1016. (**C**) Samples of HEK293T cells transiently transfected with, or without (NC), pVAX-1, pAD1002 or pAD1016 were run SDS-PAGE gel followed by Western blotting using anti-RBD^SARS^ Abs for detection. Recombinant RBD^SARS^ was included as control. (**D**) Secreted recombinant heterodimeric RBD polypeptide in culture supernatant of the transfectant HEK293T cells was quantitated using anti-RBD^SARS^ Ab-based sandwich ELISA. HRP-labeled anti-RBDWT Ab was employed as detection Ab. Serum samples, collected from BALB/c (**E**) and C57BL/6 (**F**) mice after secondary immunization with 20 μg pAD1016 DNA via MAP-1016 administration (MAP), intradermal injection (ID) or intramuscular inoculation followed by EP (IM+EP), were titrated against recombinant RBD^WT^ in ELISAs. Normal mouse serum (NMS) was included as negative control. Data represent mean ± SD (*n* = 5 biologically independent samples). The original whole blot and the densitometry readings of the bands in WB shown in Appendix A.

**Figure 2 vaccines-11-00778-f002:**
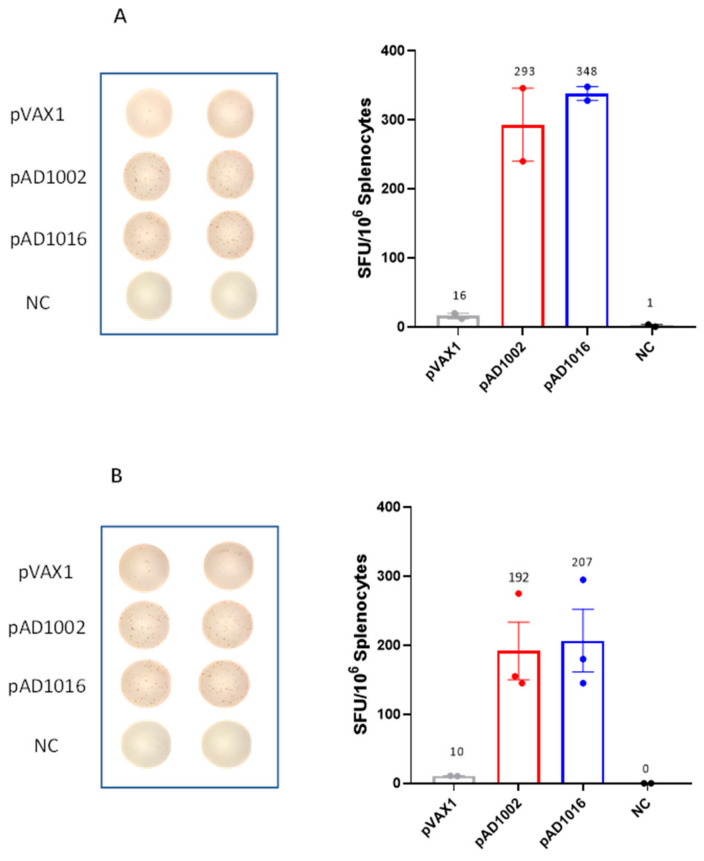
SARS-CoV-2-specific T cell responses in pAD1016-vaccinated mice. C57BL/6 mice were IM+EP immunized twice with pVAX1, pAD1002 or pAD1016 (*n* = 5, 20 μg/dose, fortnight interval) and then sacrificed for spleens 14 days after boost. ELISpot analyses of IFN-γ spot-forming cells (SFC) amongst splenocytes after re-stimulation with, or without (NC), pooled 14-mer overlapping peptides covering the sequences of RBD^SARS^ (**A**) or RBD^BA4/5^ (**B**) were performed.

**Figure 3 vaccines-11-00778-f003:**
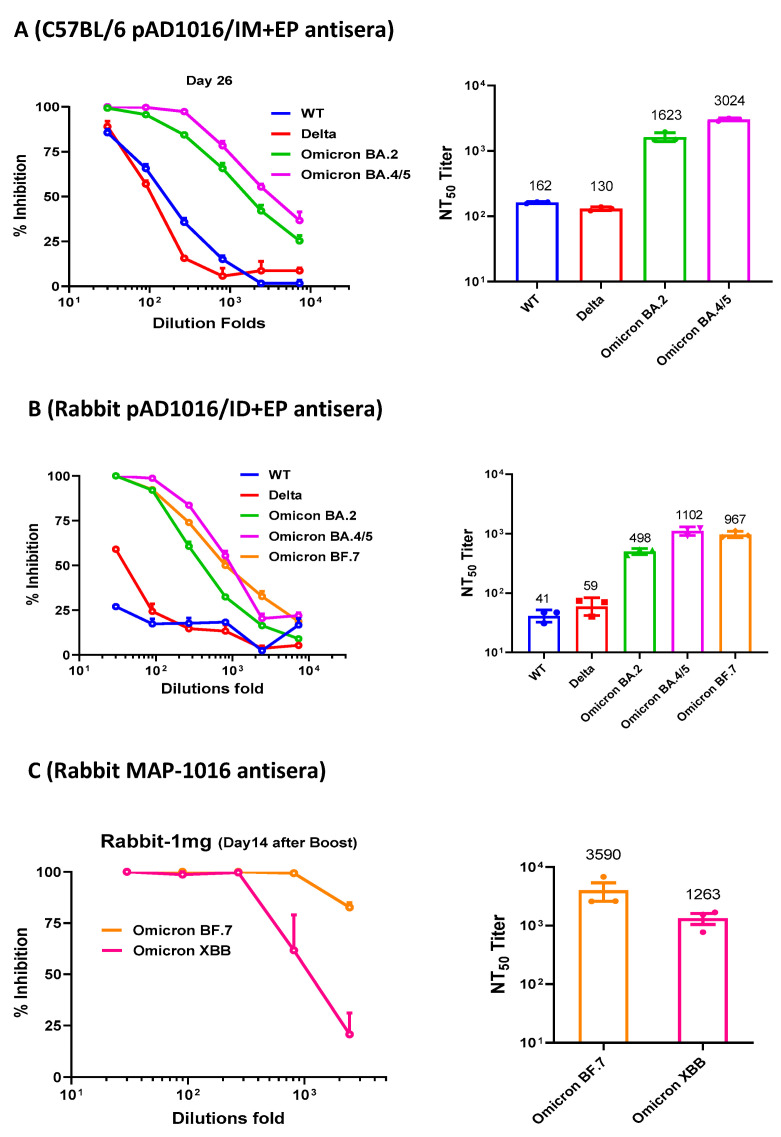
Neutralization antibodies induced by pAD1016 in mice and rabbits. (**A**) C57BL/6 mice (*n* = 5) were administered with 2 doses (20 μg/dose, fortnight interval) of pAD1016/IM+EP followed by serum sample collection 14 days after boost. An equal proportion mixture of the serum samples was tested for ability to block mimic infection of ACE2-expressing HEK293T cells by pseudoviruses displaying S protein of SARS-CoV-2 WT, Delta, Omicron BA.2 or BA.4/5 variants. (**B**) Rabbits (*n* = 3) administered with 2 doses (500 μg/dose, fortnight interval) of pAD1016/IM+EP followed by serum sample collection 14 days after boost immunization. An equal proportion mixture of the serum samples was tested for ability to block mimic infection of ACE2-expressing HEK293T cells by pseudoviruses displaying S protein of SARS-CoV-2 WT, Delta, Omicron BA.2, BA.4/5 or BF.7. (**C**) A serum sample from rabbits (equal proportion mixture of 3 rabbits) 14 days after 2 doses of MAP-1016 immunization (500 μg/MAP/dose, fortnight interval) was assayed for ability to block mimic infection of ACE2-expressing HEK293T cells by pseudoviruses displaying S protein of SARS-CoV-2 Omicron BA.7 or XBB. The results are expressed as percent inhibition of infection (left panels) and NT50 titers (right panels). Data are means ± SEM.

**Figure 4 vaccines-11-00778-f004:**
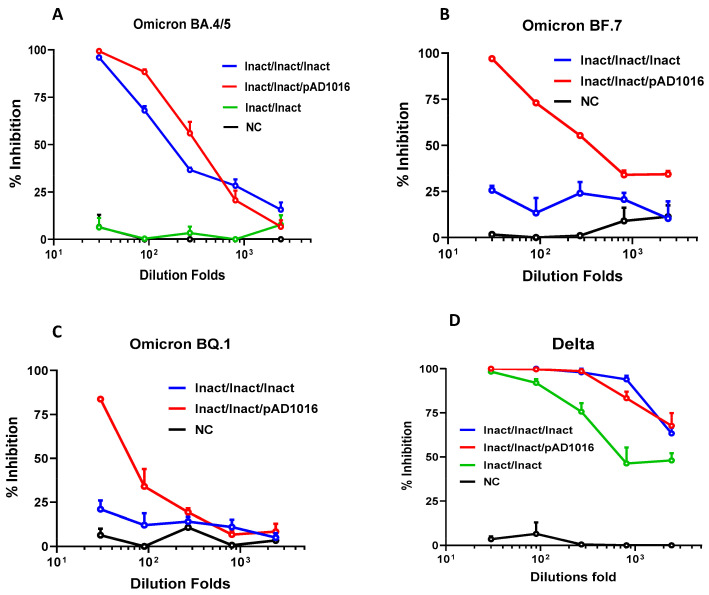
Effect of pAD1016 boost to inactivated virus pre-vaccination in mice. C57BL/6 mice (*n* = 5) were i.m. administered with two doses of inactivated SARS-CoV-2 virus vaccine (60 units/dose), and then boosted with, or without (Inact/Inact), either inactivated SARS-CoV-2 virus vaccine (Inact/Inact/Inact) or 20 μg pAD1016/IM+EP (Inact/Inact/pAD1016). Serum samples, collected 14 days after the secondary, or third, immunization were assayed for ability to block mimic infection of ACE2-expressing HEK293T cells by pseudoviruses displaying S protein of SARS-CoV-2 variants Omicron BA.4/5 (**A**), BF.7 (**B**), BQ.1 (**C**) or Delta (**D**). Normal mouse serum was included as negative control (NC). The results are expressed as percent inhibition of infection. Data are means ± SEM.

**Figure 5 vaccines-11-00778-f005:**
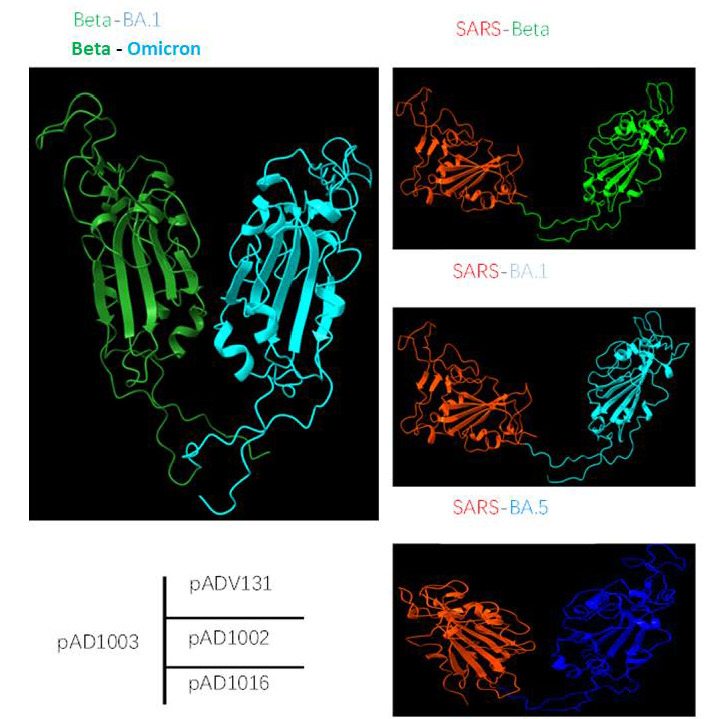
Comparison of AI-modeled molecular structure of the polypeptides encoded by pAD1002, pAD1003, pAD131 and pAD1016. Of the 4 fusion RBD heterodimers, RBD^Beta/BA1^ forms “bilateral lung” structure, indicating stable complexation between RBD^BA1^ and RBD^Beta^, while RBD^SARS/Beta^, RBD^SARS/BA1^ and RBD^SARS/BA5^ are all in free-moving conformation, suggesting a possibility that RBD^SARS^ can barely complex with RBDs of SARS-CoV-2 variants under physiological conditions.

## Data Availability

Data will be made available upon request to the corresponding author though may be subject to a Material Transfer Agreement between institutions.

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
