# Peer review of "A COVID-19 DNA Vaccine Candidate Elicits Broadly Neutralizing Antibodies against Multiple SARS-CoV-2 Variants including the Currently Circulating Omicron BA.5, BF.7, BQ.1 and XBB"

_vaccines, 2023, doi:10.3390/vaccines11040778_

Round 1

Reviewer 1 Report

  • Major comments: 

In this study, Yuan Ding et al. developed a COVID-19 DNA Vaccine, pAD1016, which encodes a receptor-binding domain (RBD) chimera of SARS-CoV-1 and Omicron BA.1. The vaccine could elicit SARS-CoV-1 and SARS-CoV-16 2 RBD-specific IFN-γ+ cellular responses in BALB/c and C57BL/6 mice. The serum Abs from vaccinated mice and rabbits could neutralize SARS-CoV-2 Omicron pseudoviruses like the strains of BA.2, BA.4/5, BF.7, BQ.1, and XBB. Besides, if used as a booster vaccine for inactivated SARS-CoV-2 virus preimmunization in C57BL/6 mice, the vaccine broadened the serum Ab neutralization spectrum to cover the Omicron BA.4/5, BF7, and BQ.1 sub-21 variants.

The study is relevant to the field and well-organized.

  • General concept comments

Here are some considerations for the study:

1.      The introduction section should discuss the advantages and disadvantages of DNA vaccines vs. mRNA vaccines, and the unique characteristics of the COVID-19 DNA vaccine candidate mentioned in this study.

2.      In Materials and Methods, please provide a figure of the cloning strategy. Moreover, the location of the RBDs used in the vaccine, the signal peptide, and linkers of the two RBD domains used, should be listed as well for better reproducibility in the field.

3.      Figure 2, the SARS-COV-2 BA.4/5 RBD peptides should be used as controls. Besides, it seems that

  • Specific comments:

1)      Line 4, “BF.5” or “BA.5”?

2)      Line 56, please give the full name of “Nabs” of its first appearance.

3)      Line 76, cDNA encoding SARS-CoV-2 variant Omicron BA.5 RBD or full length?

4)      Line 111, for qRT-PCR, what probes were used in the study?

5)      Line 115, please give the full name of “SPF” of its first appearance.

6)      Line 125, please give the full name of “IM” of its first appearance.

7)      Line 131, what kind of recombinant RBD was used for the coating of ELISA?

8)      Lines 147 and 148, what volumes of pseudovirus and sera were used, respectively, for each well?

9)      Line 151, please give the full name of “RLU” of its first appearance.

10)  Lines 155-157, “Pseudovirus neutralization experiments using Vero cells were contracted to Gobond Testing…”, what does it mean here? Using Vero cells for verification?

11)  Line 170, please give the full name of “PMA/Iono” of its first appearance.

12)  Lines 177-178, what are the continuous variables in the study?

13)  Lines 195-197, please enclose the data on human ACE2-transgenic 195 C57BL/6 and K18 mice as a supplementary file.

14)  Lines 197-199, well, the study didn’t provide evidence of the in vivo immunogenicity comparison of pAD1016 vs. pAD1002.

15)  Figure 1C, please label the size of the marker, and each lane.

16)  Lines 205/208, anti-SARS-CoV-1 RBD antibodies were used for western blot and ELISA here?

17)  Line 210, typo of “20 g pAD1016”.

18)  Lines 249-250, the rabbits were also administered with 2 doses of MAP- 1016 via MAP-mediated intradermal delivery?

19)  Lines 251-253, as there was no comparison between pAD1002 and pAD1016 in Figure 3, the description here didn’t have evidence support.

20)  Line 268, the limit of detection was 50% inhibition?

21)  Lines 304-305, please enclose the NAbs data in pigs as a supplementary file.

22)  Lines 313-315, please provide appropriate citations here to support the statement.

23)  Line 349, please give the full name of “LCs” of its first appearance.

24)  Line 353, please give the full name of “LNs” of its first appearance.

Author Response

Reviewer 1:

General concept comments

Here are some considerations for the study:

  1. The introduction section should discuss the advantages and disadvantages of DNA vaccines vs. mRNA vaccines, and the unique characteristics of the COVID-19 DNA vaccine candidate mentioned in this study.

       Following the reviewer’s suggestion, we have modified the Introduction section to discuss the advantages and disadvantages of DNA vaccines vs. mRNA vaccines (Lines 48-52). The unique characteristics of the vaccine candidate pAD1016 in inducing neutralizing Abs against recent emerging Omicron subvariants in mice, rabbits and pigs are further emphasized in this section (Lines 750-78).

  1. In Materials and Methods, please provide a figure of the cloning strategy. Moreover, the location of the RBDs used in the vaccine, the signal peptide, and linkers of the two RBD domains used, should be listed as well for better reproducibility in the field.

       A supplemental figure (Fig. S1) is added showing the cloning strategy and structural characteristics of pAD1016, together with complete amino acid sequence of the antigen encoded by pAD1016, highlighting the leader sequence and two RBD domains and clarifying that no linker sequence was added between the two RBD domains (Lines 83-84). 

  1. Figure 2, the SARS-COV-2 BA.4/5 RBD peptides should be used as controls. Besides, it seems that 

       As stated in Materials and Methods (Line 179), BA.4/5 (rather than BA1/2) RBD peptides were used in ELISpot assays. The typo in legend to Figure 2 has been corrected (Line 241).

Specific comments:

  • Line 4, “BF.5” or “BA.5”?

Corrected as BA.5.

2)      Line 56, please give the full name of “Nabs” of its first appearance.

Already given in Line 47.

  • Line 76, cDNA encoding SARS-CoV-2 variant Omicron BA.5 RBD or full length?

Rephrased as “RBD of Omicron BA.5” (Line 84).

  • Line 111, for qRT-PCR, what probes were used in the study?

  Sequences of the probes are given in the revised MS (Lines 120-121). 

  • Line 115, please give the full name of “SPF” of its first appearance.

  Done (Line 124).

  • Line 125, please give the full name of “IM” of its first appearance.

  Done (Line 135).

  • Line 131, what kind of recombinant RBD was used for the coating of ELISA?

Specified as “recombinant RBD of wildtype SARS-CoV-2 virus” in the revised MS (141).

  • Lines 147 and 148, what volumes of pseudovirus and sera were used, respectively, for each well?

For neutralization assays, 50ml pseudovirus was mixed 50ml diluted sera in microtiter plate wells. This information is given in the revised MS (Lines 157, 158).

  • Line 151, please give the full name of “RLU” of its first appearance.

The full name “relative luminance unit” for RLU is given as advised (Lines 161-162). 

  • Lines 155-157, “Pseudovirus neutralization experiments using Vero cells were contracted to Gobond Testing…”, what does it mean here? Using Vero cells for verification?

The sentence has been rephrased as “Pseudovirus neutralization results were verified using Vero cells by Gobond Testing Technology” (Lines 166-167).

  • Line 170, please give the full name of “PMA/Iono” of its first appearance.

Done (Line 180).

  • Lines 177-178, what are the continuous variables in the study?

The sentence in question with unnecessarily long statistics jargon has been rephrased (Lines 186-187).

  • Lines 195-197, please enclose the data on human ACE2-transgenic C57BL/6 and K18 mice as a supplementary file.

Supplemental Figure S2 is added showing the data on hACE2-transgenic K18 mice and pigs (Line 205).  

  • Lines 197-199, well, the study didn’t provide evidence of the in vivo immunogenicity comparison of pAD1016 vs. pAD1002.

Direct comparison between pAD1002 and pAD1016 for generation of binding and neutralizing Abs is provided in supplemental Figure S2A & 2B.

15)  Figure 1C, please label the size of the marker, and each lane.

        Missing labels for Fig. 1C have been added back.

16)  Lines 205/208, anti-SARS-CoV-1 RBD antibodies were used for western blot and ELISA here?

        For WB shown in Fig. 1C, anti-SARS-CoV-1 RBD Abs were used for detection. For EILSA shown in Fig. 1D, anti-SARS-CoV-1 RBD Abs were employed to coat the plate as capture Ab (hence anti-RBDSARS Ab-based), while HRP-labeled anti-SARS-CoV-2 RBDWT Abs were used as secondary Ab for detection (Lines 213-217).

17)  Line 210, typo of “20 g pAD1016”.

       Typo corrected.

18)  Lines 249-250, the rabbits were also administered with 2 doses of MAP- 1016 via MAP-mediated intradermal delivery? 

       Yes, the sentence has been rephrased for clearer presentation (259-263).

19)  Lines 251-253, as there was no comparison between pAD1002 and pAD1016 in Figure 3, the description here didn’t have evidence support.

       We pointed out in the Introduction section that “pAD1002 antisera were unable to neutralize Omicron subvariants BQ.1 and BF.7 pseudoviruses”, citing our previous work listed as Reference 22 (Lines 68-70). In the revised MS, we provide additional data, in supplemental Figure S2, directly comparing RBD-binding and pseudovirus blocking capability of serum Abs elicited by pAD1002 and pAD1016 in mice (Lines 252-257).

20)  Line 268, the limit of detection was 50% inhibition?

       The horizontal dash lines were used for NT50 calculation, not to show limit of detection. To avoid unnecessary confusion, the dash lines are removed from Figure 3 and Figure 4, and the sentences in question have been deleted (Lines 283 & 311).  

21)  Lines 304-305, please enclose the NAbs data in pigs as a supplementary file.

       Point taken. Pig immunization data are included in supplemental Fig. S2 (Lines 319-320; Lines 264-266). 

22)  Lines 313-315, please provide appropriate citations here to support the statement.

    In addition to citing our previous work in support of this statement, the discussion here has been expanded for clearer and better reasoning (Lines 327-334). 

23)  Line 349, please give the full name of “LCs” of its first appearance.

       “Langerhans cells” replaces “LCs” in revised text (Lines 367-368).

24)  Line 353, please give the full name of “LNs” of its first appearance.

       Already given in Line 173.

Reviewer 2 Report

The proposed modification of the vaccine produces very promising results.

 The work is important and I recommend fast publication.

Author Response

Reviewer 1:

General concept comments

Here are some considerations for the study:

  1. The introduction section should discuss the advantages and disadvantages of DNA vaccines vs. mRNA vaccines, and the unique characteristics of the COVID-19 DNA vaccine candidate mentioned in this study.

       Following the reviewer’s suggestion, we have modified the Introduction section to discuss the advantages and disadvantages of DNA vaccines vs. mRNA vaccines (Lines 48-52). The unique characteristics of the vaccine candidate pAD1016 in inducing neutralizing Abs against recent emerging Omicron subvariants in mice, rabbits and pigs are further emphasized in this section (Lines 750-78).

  1. In Materials and Methods, please provide a figure of the cloning strategy. Moreover, the location of the RBDs used in the vaccine, the signal peptide, and linkers of the two RBD domains used, should be listed as well for better reproducibility in the field.

       A supplemental figure (Fig. S1) is added showing the cloning strategy and structural characteristics of pAD1016, together with complete amino acid sequence of the antigen encoded by pAD1016, highlighting the leader sequence and two RBD domains and clarifying that no linker sequence was added between the two RBD domains (Lines 83-84). 

  1. Figure 2, the SARS-COV-2 BA.4/5 RBD peptides should be used as controls. Besides, it seems that 

       As stated in Materials and Methods (Line 179), BA.4/5 (rather than BA1/2) RBD peptides were used in ELISpot assays. The typo in legend to Figure 2 has been corrected (Line 241).

Specific comments:

  • Line 4, “BF.5” or “BA.5”?

Corrected as BA.5.

2)      Line 56, please give the full name of “Nabs” of its first appearance.

Already given in Line 47.

  • Line 76, cDNA encoding SARS-CoV-2 variant Omicron BA.5 RBD or full length?

Rephrased as “RBD of Omicron BA.5” (Line 84).

  • Line 111, for qRT-PCR, what probes were used in the study?

  Sequences of the probes are given in the revised MS (Lines 120-121). 

  • Line 115, please give the full name of “SPF” of its first appearance.

  Done (Line 124).

  • Line 125, please give the full name of “IM” of its first appearance.

  Done (Line 135).

  • Line 131, what kind of recombinant RBD was used for the coating of ELISA?

Specified as “recombinant RBD of wildtype SARS-CoV-2 virus” in the revised MS (141).

  • Lines 147 and 148, what volumes of pseudovirus and sera were used, respectively, for each well?

For neutralization assays, 50ml pseudovirus was mixed 50ml diluted sera in microtiter plate wells. This information is given in the revised MS (Lines 157, 158).

  • Line 151, please give the full name of “RLU” of its first appearance.

The full name “relative luminance unit” for RLU is given as advised (Lines 161-162). 

  • Lines 155-157, “Pseudovirus neutralization experiments using Vero cells were contracted to Gobond Testing…”, what does it mean here? Using Vero cells for verification?

The sentence has been rephrased as “Pseudovirus neutralization results were verified using Vero cells by Gobond Testing Technology” (Lines 166-167).

  • Line 170, please give the full name of “PMA/Iono” of its first appearance.

Done (Line 180).

  • Lines 177-178, what are the continuous variables in the study?

The sentence in question with unnecessarily long statistics jargon has been rephrased (Lines 186-187).

  • Lines 195-197, please enclose the data on human ACE2-transgenic C57BL/6 and K18 mice as a supplementary file.

Supplemental Figure S2 is added showing the data on hACE2-transgenic K18 mice and pigs (Line 205).  

  • Lines 197-199, well, the study didn’t provide evidence of the in vivo immunogenicity comparison of pAD1016 vs. pAD1002.

Direct comparison between pAD1002 and pAD1016 for generation of binding and neutralizing Abs is provided in supplemental Figure S2A & 2B.

15)  Figure 1C, please label the size of the marker, and each lane.

        Missing labels for Fig. 1C have been added back.

16)  Lines 205/208, anti-SARS-CoV-1 RBD antibodies were used for western blot and ELISA here?

        For WB shown in Fig. 1C, anti-SARS-CoV-1 RBD Abs were used for detection. For EILSA shown in Fig. 1D, anti-SARS-CoV-1 RBD Abs were employed to coat the plate as capture Ab (hence anti-RBDSARS Ab-based), while HRP-labeled anti-SARS-CoV-2 RBDWT Abs were used as secondary Ab for detection (Lines 213-217).

17)  Line 210, typo of “20 g pAD1016”.

       Typo corrected.

18)  Lines 249-250, the rabbits were also administered with 2 doses of MAP- 1016 via MAP-mediated intradermal delivery? 

       Yes, the sentence has been rephrased for clearer presentation (259-263).

19)  Lines 251-253, as there was no comparison between pAD1002 and pAD1016 in Figure 3, the description here didn’t have evidence support.

       We pointed out in the Introduction section that “pAD1002 antisera were unable to neutralize Omicron subvariants BQ.1 and BF.7 pseudoviruses”, citing our previous work listed as Reference 22 (Lines 68-70). In the revised MS, we provide additional data, in supplemental Figure S2, directly comparing RBD-binding and pseudovirus blocking capability of serum Abs elicited by pAD1002 and pAD1016 in mice (Lines 252-257).

20)  Line 268, the limit of detection was 50% inhibition?

       The horizontal dash lines were used for NT50 calculation, not to show limit of detection. To avoid unnecessary confusion, the dash lines are removed from Figure 3 and Figure 4, and the sentences in question have been deleted (Lines 283 & 311).  

21)  Lines 304-305, please enclose the NAbs data in pigs as a supplementary file.

       Point taken. Pig immunization data are included in supplemental Fig. S2 (Lines 319-320; Lines 264-266). 

22)  Lines 313-315, please provide appropriate citations here to support the statement.

    In addition to citing our previous work in support of this statement, the discussion here has been expanded for clearer and better reasoning (Lines 327-334). 

23)  Line 349, please give the full name of “LCs” of its first appearance.

       “Langerhans cells” replaces “LCs” in revised text (Lines 367-368).

24)  Line 353, please give the full name of “LNs” of its first appearance.

       Already given in Line 173.

Reviewer 3

Investigators of this report have constructed pAD1016 as DNA vaccine against SARS-CoV2 variants and shown that the DNA vaccination in mice and rabbits generated serum Abs capable of neutralizing multiple SARS-CoV-2 Omicron subvariants including BA.2, BA.4/5, BF.7, BQ.1 and XBB. Their data indicated the potential clinical application of pAD1016 and the worthy further translational study. The paper is well written. Minor comments:

  1. Figure 1 C lacks labels. 

Missing labels for Fig. 1C have been added back.

  1. Figure 2A and 2B are not clearly showing the results and need to be improved.   

Fig. 2 has been rearranged for improved presentation.

Reviewer 3 Report

Investigators of this report have constructed pAD1016 as DNA vaccine against SARS-CoV2 variants and shown that the DNA vaccination in mice and rabbits generated serum Abs capable of neutralizing multiple SARS-CoV-2 Omicron subvariants including BA.2, BA.4/5, BF.7, BQ.1 and XBB. Their data indicated the potential clinical application of pAD1016 and the worthy further translational study. The paper is well written. Minor comments:

1. Figure 1 C lacks labels.

2. Figure 2 A and B are not clearly showing the results and need to be improved.   

Author Response

Reviewer 3

Investigators of this report have constructed pAD1016 as DNA vaccine against SARS-CoV2 variants and shown that the DNA vaccination in mice and rabbits generated serum Abs capable of neutralizing multiple SARS-CoV-2 Omicron subvariants including BA.2, BA.4/5, BF.7, BQ.1 and XBB. Their data indicated the potential clinical application of pAD1016 and the worthy further translational study. The paper is well written. Minor comments:

  1. Figure 1 C lacks labels. 

Missing labels for Fig. 1C have been added back.

  1. Figure 2A and 2B are not clearly showing the results and need to be improved.   

Fig. 2 has been rearranged for improved presentation. 

Round 2

Reviewer 1 Report

I think that the manuscript has been improved, and the authors have addressed most of my concerns.

Here are two minor points that are needed to be addressed:

1)    Figure 1C, please label the size of the marker, and each lane.

2)    The dash lines are not removed from Figure 3 and Figure 4.

Author Response

Thank you very much for your kind comment.

(1) We now provide a new Fig. 1C with the size markers properly labeled.  

(2) The dash lines have been removed from Figs. 3 & 4.